# Shed Roof Air Extractors and Collectors: Design Guidelines for Natural Ventilation in Generic Models

**Marieli Azoia Lukiantchuki** [1],* , **Alessandra Rodrigues Prata Shimomura** [2], **Fernando Marques da Silva** [3] **and Rosana Maria Caram** [4]

1 Department of Architecture and Urbanism, State University of Maringá, Maringá 87020-900, Brazil
2 Faculty of Architecture and Urbanism, University of São Paulo, São Paulo 05508-900, Brazil
3 National Laboratory of Civil Engineering, 1700-066 Lisbon, Portugal
4 Institute of Architecture and Urbanism, University of São Paulo, São Paulo 13566-590, Brazil
* Correspondence: malukiantchuki2@uem.br

**Abstract:** Most of the Brazilian territory is classified as a hot and humid climate, whose natural ventilation is one of the most important passive design strategies. The use of this strategy can be enhanced in the design through the shed roof air collectors or extractors. However, this strategy is not exploited by architecture design, due to the designers' lack of knowledge about the efficiency of these devices. The article's aim is to present guidelines for the design of shed roof air extractors and collectors, seeking to help designers to use these devices in buildings. The method is parametric studies, through CFD simulations. For the shed roof air extractors and collectors, the following is recommended: aerodynamic geometries; building with less depth and large air outlet openings. The increase in the number of sheds influences ventilation more than the change in the geometry of the sheds. For extraction, the area of the air outlet openings is the parameter that exerts the greatest influence on ventilation. For collection, the increase in the sizes of the openings of the sheds, without changing other parameters, does not significantly increase the air speed.

**Keywords:** shed roofs; natural ventilation; design guidelines; CFD simulation



## 1. Introduction

Natural ventilation in buildings is highly associated with air quality, human health and user productivity. According to Royan et al. (2018) [1] and Mundhe et al. (2018) [2], the composition of indoor air is altered by human activities and by the emission of volatile organic compounds by the materials that make up the buildings. Considering that, on average, people stay from 80% to 90% of the time in buildings, it is important to provide good indoor air quality, through renovation, assisting in the dispersion of pollutants, odor removal, among other microorganisms.

The quality of indoor air becomes even more significant at the current time of the COVID-19 pandemic. Research shows that closed environments and under certain conditions—poor ventilation, long-term activities, users speaking out loud or doing heavy exercises—enable highly virulent and dominant airborne transmission for the spread of SARS-CoV-2 [3,4]. In addition, according to Prather et al. (2020) [5] and Tellier et al. (2019) [6], airborne transmission may also be partially responsible for the high secondary transmission rates for medical staff, as well as outbreaks in nursing facilities. According to Godwin et al. (2021) [7], experiments in both fluid mechanics and microbiology provided evidence that transmission of the virus through aerosols is possible. Different pieces of current research have highlighted the use of natural ventilation in indoor environments, as part of a set of actions to face the COVID-19 pandemic [8–11].

In addition to the quality of indoor air, natural ventilation is an efficient strategy for obtaining thermal comfort from users passively, especially in regions with hot and humid climates, such as most of the Brazilian territory [12]. In these regions, air currents increase

heat exchanges by convection, providing physiological cooling. The effectiveness of this cooling depends on the speed and temperature of the air, so that high temperature values can be reduced by increasing this speed. However, there is an optimum air speed, and after this value, an increase has little effect on the thermal sensation of the user. This ideal speed is not constant and depends on temperature, relative air humidity, metabolic rate and clothing of users [13].

Natural ventilation is the most important design strategy for the Brazilian reality, after shading, and, according to Lamberts et al. (2014) [14], most Brazilian capitals demand its use as the main strategy for the summer and even throughout the year. However, despite the fact that most of the Brazilian territory is classified as a hot humid climate, Brazil, due to its immense territory and the fact that it is located between two tropics, has a very varied climate. Considering the local needs of each region of the country, natural ventilation has been shown to be an applicable solution, and its control and/or increment can be done through design decisions, seeking to adapt a design to different regions. Both the direction and the speed of the external winds of a given region cannot be changed directly by the designers. However, the designer can act taking the best advantage of these characteristics through design solutions, such as building implementation, sizes of the openings and the use of strategies such as solar chimney [15–17], air collectors [18,19], ventilated sills [20], sheds [21–24], among others.

Among the design strategies for natural ventilation, shed roofs stand out, which are the focus of this article. Sheds are solutions at the level of coverage that can be designed for both extraction and air collection, depending on the orientation of their openings in relation to the prevailing winds: (1) in extractor sheds, the roof openings are oriented against the prevailing winds (negative pressure region—leeward), work as an air outlet, and the collection takes place on the façade in areas of positive pressure or windward; (2) in the collector sheds, the roof openings are oriented in the direction of the prevailing winds (positive pressure region—windward), and the air outlet is on a lower level, on the leeward façade (Figure 1). Compared to flat roofs, sheds admit natural ventilation and lighting, significantly contributing to the thermal comfort of users and healthier indoor environments.

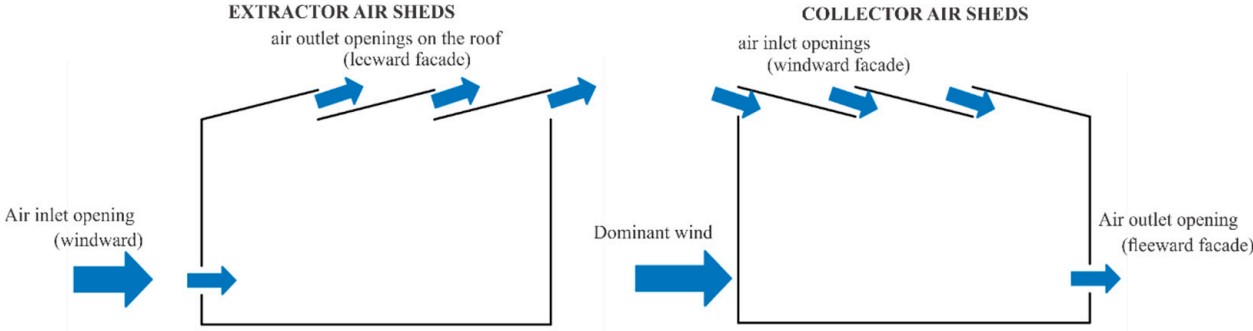

**Figure 1.** Natural ventilation through sheds, in orthogonal format, air extractors and collectors.

The most common examples of these devices are the so-called "saw tooth", whose shape is orthogonal and without many variations (Figure 1). In Brazil, the work of the architect João Filgueiras Lima, also known as Lelé, stands out, whose shed roof air extractors and collectors are in most of his designs and with different aerodynamic geometries (Figure 2), seeking to improve natural ventilation and lighting. The concern with the format of these devices is important because for Hoof et al. (2011), one of the main parameters that influence natural ventilation is the building's geometry. Internationally, studies have been developed for orthogonal shaped sheds [25–31]. Few studies address these devices with aerodynamic shapes [24,32], with a significant part being developed by the authors of this article.

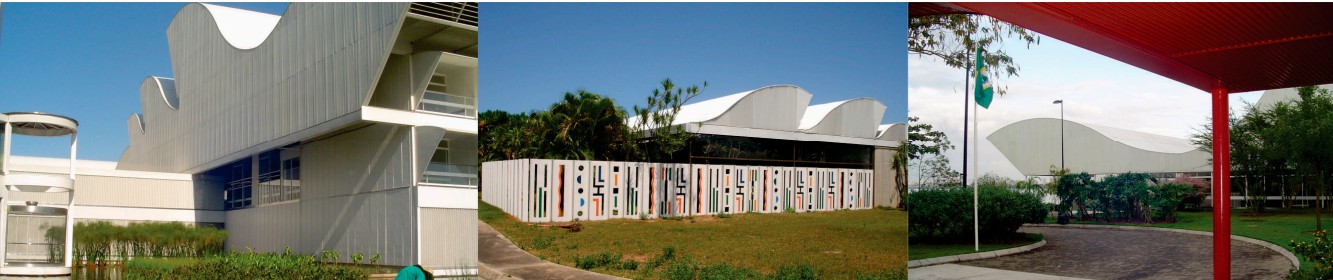

**Figure 2.** Aerodynamic sheds by architect Lelé.

Previous publications made by the authors pointed out the best and worst parameters for the shed roof air extractors and collectors design [21,22]. From these works, it can be concluded (for both types of sheds): (1) the aerodynamic model allows better internal air circulation conditions, considerably increasing the speed of the internal airflow [32]; (2) performance gains were registered with the increase in the areas of the air outlet openings. It is noteworthy that the increase in the air inlet opening is also an important factor for air extractors. In the case of collector sheds, the isolated increase in their inlet openings is influenced by other factors, such as, for example, a shorter distance between sheds located at the same height. This makes it impossible for air to enter through the devices located in the posterior region and, thus, there is a reduction in air speed in indoor environments [32]; (3) the increase in the number of sheds significantly reduced the rates of air renewal in the indoor environment, which is even more intense for sheds as collectors [22]; (4) in extractor sheds, the increase in the distance between these devices significantly reduced the renewal rates, mainly due to the reduction of the total air outlet area. On the contrary, for collector sheds, a slight increase in the internal airflow was noted, because, although the increase in distance reduced the number of air inlet openings, it made the posterior sheds more effective for collecting air [21]; (5) the collection of the airflow through the roof becomes even more effective if, together with the increase in the distance between these elements, the sheds were arranged in a misaligned way [21].

Despite the potential of sheds to increase natural ventilation in indoor spaces, these devices have not been explored much by architecture. This is due to the lack of knowledge of designers about the efficiency of these devices for natural ventilation. Most publications on the performance of design strategies, especially sheds, have a scientific format making it difficult for designers to use this data when designing buildings. Despite the scientific publications mentioned above [21,22,32], present qualitative and quantitative data regarding air extractor and collector sheds, the results are not clearly presented to designers. Note the lack of publications with design guidelines directed to the application of sheds in architectural projects. Thus, the objective of the article is to present the guidelines and recommendations for the shed roof air extractors and collectors design, in generic models, based on the scientific discussions held in previous publications.

## 2. Materials and Methods

The method was based on a parametric analysis that consists of the variation of design parameters, to verify their influences on the performance of the shed roof air extractors and collectors. The steps are described in detail below.

### 2.1. Selection of Cases Analysed

First, the reference case was defined, which will be the basis for the analysis of the different design parameters. The characteristics of the buildings designed by Lelé were used to define this case, because he is known worldwide for the use of sheds to promote natural ventilation and because the most of his buildings are located in regions with hot and humid climates [32]. The geometry of the shed in the reference case has an orthogonal shape, as it is the format used in most standard buildings with shed roofs (Figure 3). The

focus was to analyze the influence of different design parameters on the performance of the shed roof air extractors and collectors. However, the importance of evaluating, in future research, the insertion of protectors in the openings to avoid the incidence of direct solar radiation and the entry of undesirable elements (such as rain), is emphasized.

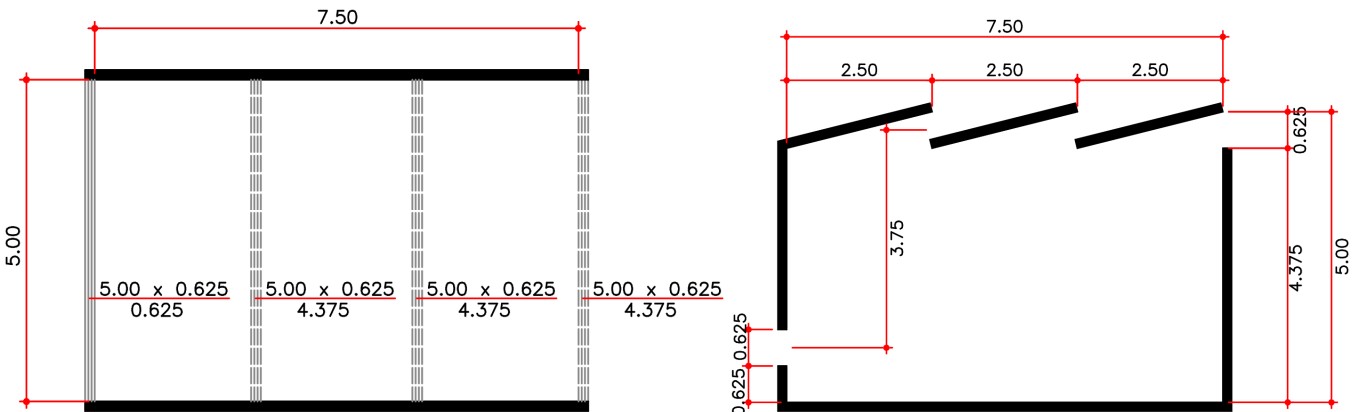

**Figure 3.** Reference case with respective sizes.

The analyzed parameters were organized in 3 groups: (1) sheds' geometry [26], where in addition to the reference case, two other geometries were analyzed; (2) variation in the sizes of the openings for cases of less depth [32]; (3) variation in the sizes of the openings, the distance and the alignment between the sheds for the models of greater depth [21,22]. The choice for these parameters was due to the influence they have on the shed roof air extractors and collectors, as described in the introduction. Table 1 presents the geometries of the sheds analyzed with the dimensions of all design parameters. Table 2 presents the variation of air inlet and outlet openings, in the three geometries of the sheds analyzed, in cases with 3 sheds. Lastly, Table 3 presents the variations in opening and distance and alignment of the shed, in three geometries of the sheds analyzed, in cases with 7 sheds. The highlighted parameters are those that have changed for the evaluation.

**Table 1.** Cases analyzed in group 01: geometry of the sheds.

| Analyzed Geometry | |
|---|---|
| Geometry One (G01) | Geometry Two (G02) |
| 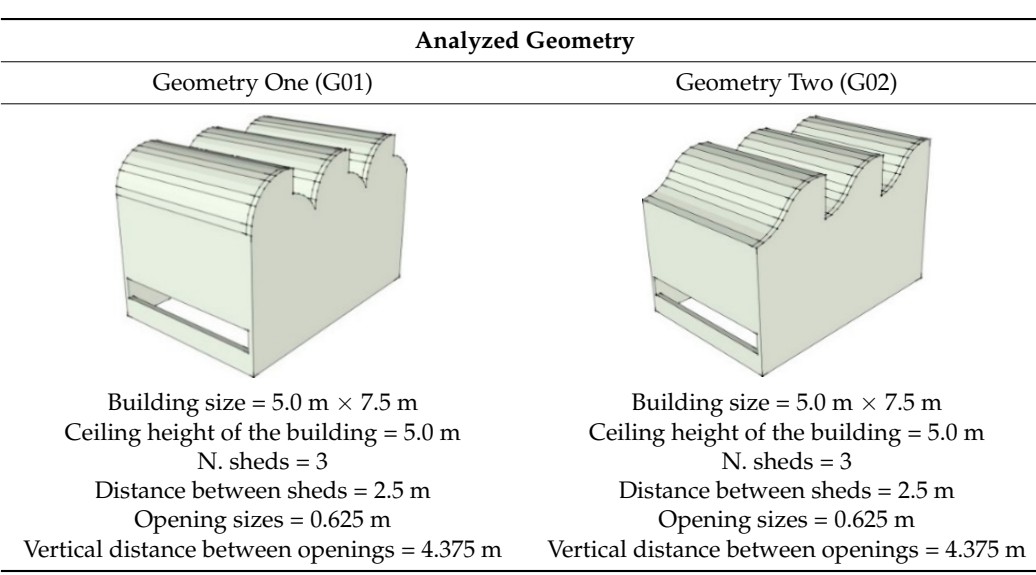 | |
| Building size = 5.0 m × 7.5 m | Building size = 5.0 m × 7.5 m |
| Ceiling height of the building = 5.0 m | Ceiling height of the building = 5.0 m |
| N. sheds = 3 | N. sheds = 3 |
| Distance between sheds = 2.5 m | Distance between sheds = 2.5 m |
| Opening sizes = 0.625 m | Opening sizes = 0.625 m |
| Vertical distance between openings = 4.375 m | Vertical distance between openings = 4.375 m |

**Table 2.** Cases analyzed in group 02: variation of air inlet and outlet openings.

| Cases | Reference Case | Geometry One | Geometry Two |
|---|---|---|---|
| **CA** |  |  |  |
| | N. sheds = 03 | N. sheds = 03 | N. sheds = 03 |
| | Opening height = 0.625 m | Opening height = 0.625 m | Opening height = 0.625 m |
| | Opening area = 3.12 m$^2$ | Opening area = 3.12 m$^2$ | Opening area = 3.12 m$^2$ |
| | Sheds height = 0.625 m | Sheds height = 0.625 m | Sheds height = 0.625 m |
| | Sheds Total Area = 9.375 m$^2$ | Sheds Total Area = 9.375 m$^2$ | Sheds Total Area = 9.375 m$^2$ |
| **CA1** |  |  |  |
| | N. sheds = 03 | N. sheds = 03 | N. sheds = 03 |
| | Opening height = 0.9375 m | Opening height = 0.9375 m | Opening height = 0.9375 m |
| | Opening area = 4.68 m$^2$ | Opening area = 4.68 m$^2$ | Opening area = 4.68 m$^2$ |
| | Sheds height = 0.625 m | Sheds height = 0.625 m | Sheds height = 0.625 m |
| | Sheds Total Area = 9.375 m$^2$ | Sheds Total Area = 9.375 m$^2$ | Sheds Total Area = 9.375 m$^2$ |
| **CA2** |  |  |  |
| | N. sheds = 03 | N. sheds = 03 | N. sheds = 03 |
| | Opening height = 1.25 m | Opening height = 1.25 m | Opening height = 1.25 m |
| | Opening area = 6.25 m$^2$ | Opening area = 6.25 m$^2$ | Opening area = 6.25 m$^2$ |
| | Sheds height = 0.625 m | Sheds height = 0.625 m | Sheds height = 0.625 m |
| | Sheds Total Area = 9.375 m$^2$ | Sheds Total Area = 9.375 m$^2$ | Sheds Total Area = 9.375 m$^2$ |
| **CS1** |  |  |  |
| | N. sheds = 03 | N. sheds = 03 | N. sheds = 03 |
| | Opening height = 0.625 m | Opening height = 0.625 m | Opening height = 0.625 m |
| | Opening area = 3.12 m$^2$ | Opening area = 3.12 m$^2$ | Opening area = 3.12 m$^2$ |
| | Sheds height = 0.937 m | Sheds height = 0.937 m | Sheds height = 0.937 m |
| | Sheds Total Area = 14.062 m$^2$ | Sheds Total Area = 14.062 m$^2$ | Sheds Total Area = 14.062 m$^2$ |

**Table 2.** *Cont.*

| Cases | Reference Case | Geometry One | Geometry Two |
|---|---|---|---|
| CS2 |  |  |  |
| | N. sheds = 03 | N. sheds = 03 | N. sheds = 03 |
| | Opening height = 0.625 m | Opening height = 0.625 m | Opening height = 0.625 m |
| | Opening area = 3.12 m$^2$ | Opening area = 3.12 m$^2$ | Opening area = 3.12 m$^2$ |
| | Sheds height = 1.25 m | Sheds height = 1.25 m | Sheds height = 1.25 m |
| | Sheds Total Area = 18.75 m$^2$ | Sheds Total Area = 18.75 m$^2$ | Sheds Total Area = 18.75 m$^2$ |

**Table 3.** Cases analyzed in group 03: opening, distance and alignment of the sheds.

| Cases | Reference Case | Geometry One | Geometry Two |
|---|---|---|---|
| CS7 |  |  |  |
| | Model size = 5.0 m × 17.5 m | | |
| | N. sheds = 07 | | |
| | Distance between sheds = 2.50 m | | |
| | Sheds height = 0.625 m | | |
| | Sheds Total Area = 21.875 m$^2$ | | |
| CS8 |  |  |  |
| | Model size = 5.0 m × 17.5 m | | |
| | N. sheds = 05 | | |
| | Distance between sheds = 3.75 m | | |
| | Sheds opening height = 0.625 m | | |
| | Sheds Total Area = 15.325 m$^2$ | | |
| CS9 |  |  |  |
| | Model size = 5.0 m × 17.5 m | | |
| | N. sheds = 04 | | |
| | Distance between sheds = 5.00 m | | |
| | Sheds opening height = 0.625 m | | |
| | Sheds Total Area = 12.5 m$^2$ | | |

**Table 3.** *Cont.*

| Cases | Reference Case | Geometry One | Geometry Two |
|---|---|---|---|
| **CS10** |  |  |  |
| | Model size = 5.0 m × 17.5 m | | |
| | N. sheds = 03 | | |
| | Distance between sheds = 7.50 m | | |
| | Sheds opening height = 0.625 m | | |
| | Sheds Total Area = 9.375 m$^2$ | | |
| **CS11** |  |  |  |
| | Model size = 5.0 m × 17.5 m | | |
| | N. sheds = 02 | | |
| | Distance between sheds = 15.0 m | | |
| | Sheds opening height = 0.625 m | | |
| | Sheds Total Area = 6.25 m$^2$ | | |
| **CS12** |  |  |  |
| | Model size = 5.0 m × 17.5 m | | |
| | N. sheds = 04 | | |
| | Distance between sheds = 5.00 m | | |
| | Sheds opening height = 1.0937 m | | |
| | Sheds Total Area = 21.875 m$^2$ | | |
| **CS13** |  |  |  |
| | Model size = 5.0 m × 17.5 m | | |
| | N. sheds = 03 | | |
| | Distance between sheds = 7.50 m | | |
| | Sheds opening height = 1.4583 m | | |
| | Sheds Total Area = 21.875 m$^2$ | | |

**Table 3.** *Cont.*

| Cases | Reference Case | Geometry One | Geometry Two |
|---|---|---|---|
| **CAD1** |  |  |  |
| | Model size = 5.0 m $\times$ 17.5 m | | |
| | N. sheds = 07 | | |
| | Distance between sheds = 2.50 m | | |
| | Sheds opening height = 0.625 m | | |
| | Sheds Total Area = 10.9375 m$^2$ | | |
| **CAD2** |  |  |  |
| | Model size = 5.0 m $\times$ 17.5 m | | |
| | N. sheds = 04 | | |
| | Distance between sheds = 5.00 m | | |
| | Sheds opening height = 0.625 m | | |
| | Sheds Total Area = 6.25 m$^2$ | | |
| **CAD3** |  |  |  |
| | Model size = 5.0 m $\times$ 17.5 m | | |
| | N. sheds = 03 | | |
| | Distance between sheds = 7.50 m | | |
| | Sheds opening height = 0.625 m | | |
| | Sheds Total Area = 4.6875 m$^2$ | | |
| **CAD4** |  |  |  |
| | Model size = 5.0 m $\times$ 17.5 m | | |
| | N. sheds = 06 | | |
| | Distance between sheds = 5.00 m | | |
| | Sheds opening height = 0.625 m | | |
| | Sheds Total Area = 6.25 m$^2$ | | |

### 2.2. Selection of Climatic Data

First, the wind speed values of cities located in the Bioclimatic Zone 8 [33] were selected through a survey of climatic data from National Meteorological Institute (Portuguese acronym: INMet) and in the epw files prepared by Roriz (2012) [34]. This selection occurred due to the recommendation to use natural ventilation throughout the year in these regions. There was a variation in the speed values between the selected cities. Thus, three speed values were defined and analyzed, being 1.5 m/s–3.0 m/s and 7.0 m/s, characterized by low, medium and high, respectively. The analyses were carried out for the isolated building, that is, without a built environment, due to the high computational demand required. However, the Atmospheric Boundary Layer (ABL) was simulated, correcting the value of this speed at the building height (Equation (1)). For the simulations, it was adopted as a suburban built environment with $\alpha = 0.21$, obtained through tests in the wind tunnel [24,35].

$$\frac{U}{U_{ref}} = \left(\frac{h}{h_{ref}}\right)^{\alpha} \tag{1}$$

where:

$U$ = average wind speed at a certain time h (m/s); $U_{ref}$ = Wind speed measured at the reference height (m/s); $h$ = building height at which the wind speed must be evaluated (m); $h_{ref}$ = wind speed reference height (10 m); $\alpha$ = power law exponent of the atmospheric boundary layer (surroundings).

The sheds were evaluated as extractors (0° and 45°) and collectors (135° and 180°) (Figure 4), analyzing the effect of natural ventilation by the action of the winds. The 90° direction has been previously analyzed and has significantly reduced natural ventilation performance. Changing the design parameters did not cause an increase in the speed of the indoor air [21] and, therefore, this direction was not analyzed in the article.

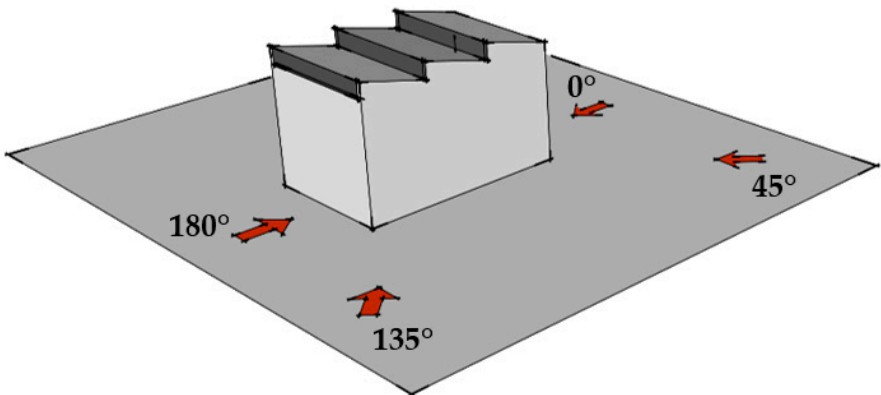

**Figure 4.** Reference case with respective sizes.

### 2.3. Computer Simulation

#### 2.3.1. Model Generation

For this step, software based on the Computational Fluid Dynamics (CFD) CFX 12.0, Ansys was used. The computational simulations were validated through wind tunnel tests in National Civil Engineering Laboratory—LNEC, Lisbon. The results were published in Lukiantchuki, et al., (2018) [22] and Lukiantchuki, et al., (2019) [32]. The sizes of the rectangular domain followed the recommendations of and Tominaga et al. (2008) [31]: distances to windward and on the sides = 5 H (25 m); height = 6 H (30 m); distance to leeward = 15 H (75 m), H = 5 m corresponding to the height of the simulated model (Figure 5).

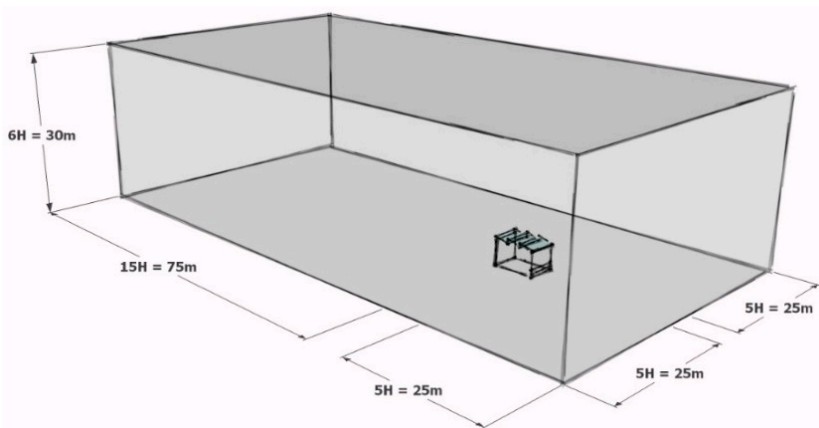

**Figure 5.** Sizes adopted for the rectangular domain.

The area of obstruction of the building in the domain was a maximum of 1.5% in the direction perpendicular to the flow, following the suggestion of Cost (2004) [36] and Tominaga et al. (2008) [37]. Thus, the blocking effect is avoided, preventing the domain boundaries from influencing the flow.

### 2.3.2. Computational Mesh Generation

For all simulations, a structured tetrahedral mesh was used, whose parameters adopted were: maximum element size16; natural size 4; cells in gap 8. The mesh was refined in the building at 0.20 m, to improve the visualization of the airflow inside. Tominaga et al. (2008) [37] points out that predicting the flow around the building with high precision is important to correctly reproduce the flow separation near the roof and walls. Thus, it is necessary to refine the mesh in the areas of interest. Inflation parameters were set for the complex geometry face elements (sheds) to generate a finely resolved mesh normal to the wall and coarse parallel to it which resolve the boundary layer properly at relatively less computational cost (Figure 6). The combination of these parameters determines the number of elements and, thus, the processing time of the simulations. These parameters were defined based on mesh sensitivity tests, which were published in Lukiantchuki (2018) [35].

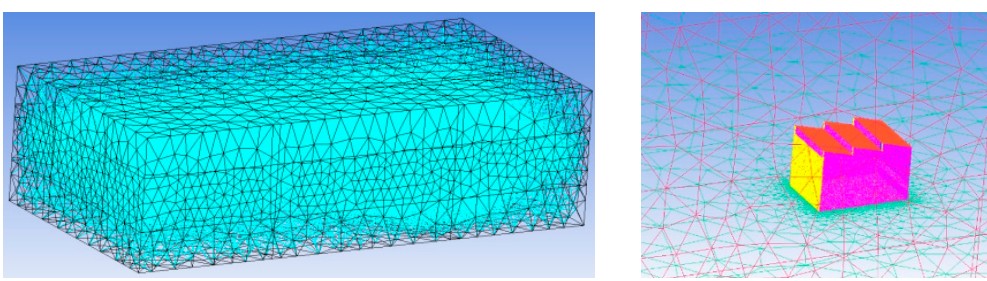

**Figure 6.** Details of the mesh generated for the reference case.

### 2.3.3. Defining the Initial and Boundary Conditions

The definition of the turbulence model, the initial conditions and the boundary conditions was based on Cóstola and Alucci (2007) [38] and Tominaga et al. (2008) [37]. The domain conditions were defined similarly to those of a wind tunnel, considering that the simulations were verified through experiments [32]: inlet and outlet as domain surfaces for air intake and outlet, respectively and the lateral faces, floor and ceiling as walls, not allowing airflow to pass through. The simulation was carried out in a steady state, in the isothermal condition at 25 °C. For the assessment of natural ventilation by the action of winds, the same ABL profile obtained by tests in the wind tunnel [32] was used as an

inlet condition in the CFD simulations. The same experimental conditions were adopted: Uref = 7 m/s and $\alpha$ = 0.21, based on suburban environmental surroundings and to generate the wind profile Equation (1) was applied (Figure 7). The effect produced by the different characteristics of the surrounding environment significantly influences in the distribution of airflow inside the building and the values of Cp. Therefore, wind data at the level of the building must be correct and an atmospheric boundary layer must be generated.

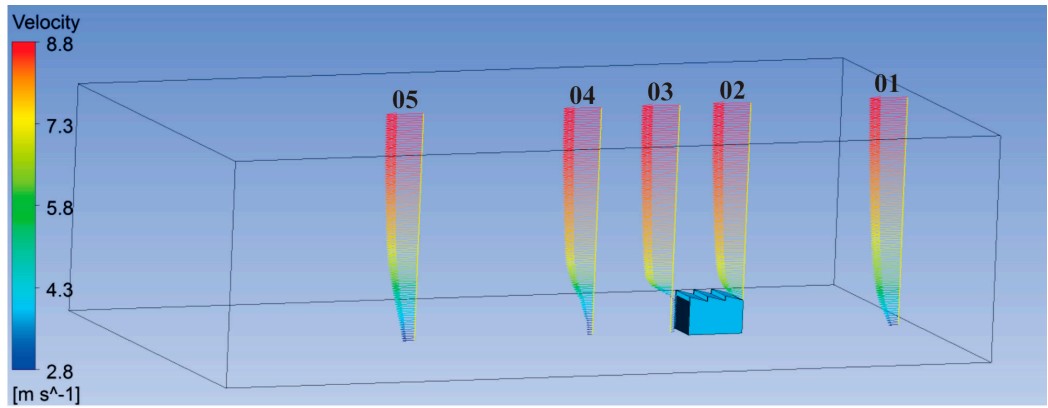

**Figure 7.** Profile of Atmospheric Boundary Layer in CFD simulation.

The turbulence model used was K-epsilon, which is well established in several studies of fluid dynamics and natural ventilation [39,40]. Cóstola and Alucci (2007) [38], Cóstola et al. (2009) [39] and Cost (2004) [37] recommend the use of the DNS (Direct Numerical Simulation) or LES (Large Eddy Simulation), which have superior reliability than other models. However, the computational capacity must be large, which makes the simulations difficult. In addition, the model adopted is well established by several studies of natural ventilation [39,40]. The low computational demand of the k-Epsilon model makes it one of the most validated and applied models in engineering. Second-order models have high computational demands, in addition to increasing the processing time by more than three times, which limits their application [41,42]. It should be noted that simulations with other turbulence models would be important in future research, as it was not the focus of the article. The level of convergence was established when all residual levels reached a maximum value of 10–4. The minimum number of iterations was 1000.

### 2.4. Analysis Parameters

Due to the COVID-19 pandemic, the parameter of air renewal rate per hour gains significant importance, aiming to assign recommended values for renewal of indoor air, depending on the activities developed and the area of the environment. At this time, the hygienic function of natural ventilation is enhanced, helping to reduce the transmission of pathogens. However, the standards present values that have been discussed, which can, consequently, result in the revision of the minimum values for maintaining indoor air quality. Considering that the focus of the article is for hot and humid climates and the lack of current comparative values on air renewal rates for this climatic reality, the average indoor air velocity was used as an analysis parameter. In addition, it is assumed that models that meet the parameters of thermal comfort of users also meet hygienic issues, because, according to the regulations, they are low values.

Therefore, the analysis was carried out aiming at the comfort of users for regions with hot and humid climate, considering that most of the Brazilian territory is classified in this way, and because they are the places that have a great need for ventilation for thermal comfort. First, the ventilation performance was analyzed by the shed roof air extractors and collectors, through the percentage parameter of utilization of the average speed of the indoor airflow ($U$), as a function of the external wind speed ($U_{ref}$). Then, the effect on the thermal comfort of the users was analyzed through the average speed inside the models, depending on the

winds. Generally, acceptable airspeed limits are provided by standards [43–45], which specify values for airspeed lower than those desired by users in regions of Brazil that require greater air movement. Thus, the values of average air velocity in the internal environment were obtained, as a function of wind speeds, and the limits proposed by Cândido et al. (2010) [12], who questioned the air acceptability limits foreseen by the referred standards.

Experiments in Brazil have shown that our limits are different from those practiced in European countries, because, in hot climates, the movement of air that is considered uncomfortable in cold and temperate climates, may be welcome to users in relation to thermal comfort. Cândido et al. (2010) [12] noted that higher speeds are accepted and even desired by users. Table 4 presents the scale elaborated based on the research of Cândido et al. (2010) [12], which expands the maximum limits of the average air velocity in indoor spaces.

**Table 4.** Limits of acceptability of air speed in indoor environments sheds.

| Air Velocity (m/s) | Situation Occurred |
| --- | --- |
| 0–0.2 | Imperceptible natural ventilation |
| 0.2–0.4 | Perceptible natural ventilation |
| 0.4–0.8 | Satisfactory natural ventilation (there is a reduction in thermal load and contributes to comfort) |
| Above 0.8 | Control is necessary (discomforts such as lifting papers, objects disorder) |

## 3. Results

### 3.1. Design Guideline for the Use of the Shed Roof Air Extractors and Collector

Figure 8 shows the use of wind in the internal environment ($U/U_{ref}$) for the cases of group 1, for the shed roof air extractors and collectors. Table 5 gathers the design guidelines, with respect to the geometry parameter of the sheds, for the same situations. The red highlight punctuates the situation whose performance was superior.

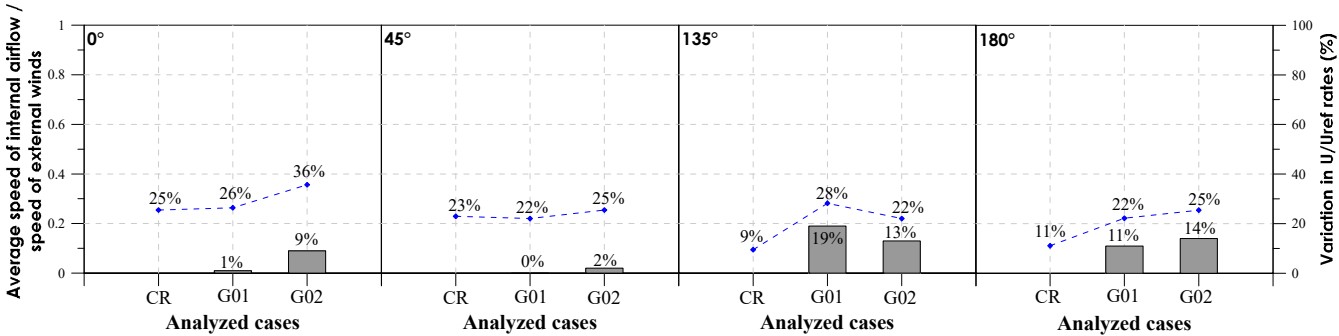

**Figure 8.** Percentage of utilization of the average speed of the indoor airflow as a function of the speed of the external winds ($U/U_{ref}$) for group 1: geometry of the sheds.

**Table 5.** Design guidelines for group 1: sheds 'geometry, for air collection and extraction.

| Design Parameter 01: SHEDS GEOMETRY | | |
| --- | --- | --- |
| Air EXTRACTOR Sheds | Shed Roof Air EXTRACTORS and COLLECTORS | Shed Roof Air EXTRACTORS and COLLECTORS |
| G02 | G01     G02 | G02 |

**Table 5.** *Cont.*

| Design Parameter 01: SHEDS GEOMETRY |
| --- |
| Based on Lukiantchuki et al. (2016) [32], we have the best performance: |

- Air extractor sheds: G02 showed the use of external winds around 10% higher than the CR and G01, for winds affecting perpendicularly to the openings (0°). It is noted that when the incident external wind is at 45°, the performance between the geometries is very similar, showing variations of around 1%. Thus, it is noted that the wind incidence angle (0 and 45) for the shed roof air extractors did not have a significant difference in the natural ventilation performance.
- Air collector sheds: G01 and G02 presented themselves as good options, presenting similar performances (22% and 25%, respectively).
- Shed roof air extractors and collectors: G02 becomes interesting for the reality of most urban environments, whose wind direction varies throughout the day and year (showed an improvement of 36% and 25% for the air extractors and collectors sheds, respectively).
- In all the geometries analyzed, the sheds as air extractors had a better performance than the collectors (in the case of the G02, the utilization was 11% higher when the wind falls at 0°). This occurred because, in the case of collection, regardless of the geometry, the air is directed to the interior of the building only by the first shed, due to the wind shadow on the posterior sheds, caused by the air current that touches the roof.
- Regarding this parameter, it is emphasized that it is not possible to issue an accurate recommendation, considering that changes in geometry can present significant changes in the internal airflow. Thus, it is recommended to carry out assessments with a wind tunnel or CFD simulations for complex geometries.

Figure 9 shows the use of external wind in the internal environment (U/Uref) for the cases of group 2, for the shed roof air extractors and collectors. Then, Table 6 gathers the design guidelines, regarding the size parameter of the air inlet and outlet openings, for the same situations.

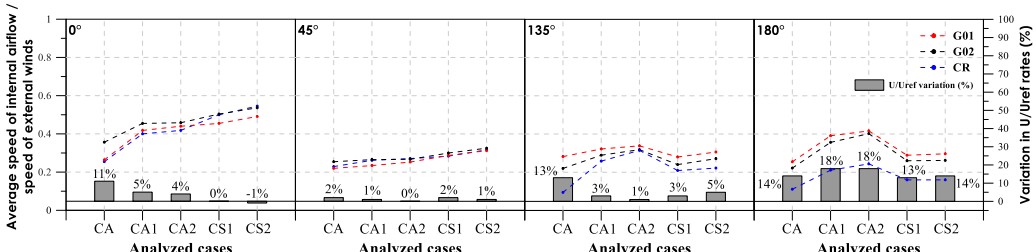

**Figure 9.** Percentage of utilization of the average speed of the indoor airflow as a function of the speed of the external winds ($U/U_{ref}$) for group 2: size of the air inlet and outlet openings.

Figure 10 shows the use of external wind in the internal environment (U/Uref) for the parameter number of sheds on the roof, for the shed roof air extractors and collectors. Table 7 gathers the design guidelines for the number of sheds.

**Table 6.** Design guidelines for group 2: size of the air inlet and outlet openings.

| Design Parameter 02: SIZE OF AIR INLET AND OUTLET OPENINGS | | |
| --- | --- | --- |
| Air EXTRACTOR Sheds | Air COLLECTOR Sheds | Shed Roof Air EXTRACTORS and COLLECTORS |
| CS2 + G02 CA2 + G02 | CA2 + G01 CA2 + G02 | CS2 + G02 |

**Table 6.** *Cont.*

| Design Parameter 02: SIZE OF AIR INLET AND OUTLET OPENINGS |
| :--- |
| Based on Lukiantchuki et al. (2016) [32], we have the best performance:<br><br>• Air extractor sheds: air outlet openings (sheds) with larger sizes. The increase in the inlet opening also provided an increase in the speed of the indoor air (around 11%), occurring, even, situations with values higher than the collectors.<br>• Air extractor sheds: the orthogonal geometry (CR) performed well with the increase in the size of its openings (the increase for G02 was only 4% to 5% when the inlet openings were increased and the performance was the same for both geometries when the openings of the sheds were enlarged). Thus, the designer can choose to use sheds with orthogonal geometries, increasing the air outlet openings.<br>• Air collector sheds: air outlet openings (façade) with large sizes. In this case, the isolated increase in the air inlet openings (sheds) was influenced by other aerodynamic effects, such as wind shadow, due to the greater proximity of the sheds and at the same height, which prevents the inlet of air through the posterior devices and, therefore, a better performance. In other words, the increase in the sizes of the openings of the sheds, without changing other parameters, did not make the posterior openings effective for collection, thus, the increase in air velocity was not so significant (showing an increase of only 1% from CS1 to CS2).<br>• Shed roof air extractors and collectors: buildings with large inlet and outlet openings.<br>• Air extractor sheds performed better than collectors did. |

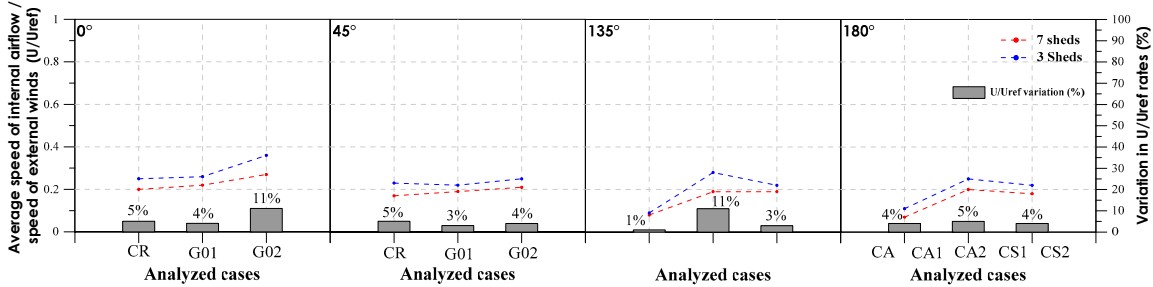

**Figure 10.** Percentage of utilization of the average speed of the indoor airflow as a function of the speed of the external winds ($U/U_{ref}$) for group 3: opening, distance and alignment of the sheds.

**Table 7.** Design guidelines for guidelines on the number of sheds on the roof.

| Design Parameter 03: NUMBER OF SHEDS IN THE ROOF | | |
| :---: | :---: | :---: |
| Air EXTRACTOR Sheds | Air COLLECTOR Sheds | Shed roof air EXTRACTORS and COLLECTORS |
| CS2 + G02 | CS2 + G01   CS2 + G02 | CS2 + G02 |
| Based on Lukiantchuki et al. (2018) [21], we have the best performance:<br><br>• Air extractor sheds: building with less depth. Although the increase in sheds in the roof increased the number of outlet openings, the speed of the airflow in the internal space was reduced for all analyzed geometries. The isolated increase in this design parameter reduced the difference in pressure coefficients between the air inlet and outlet openings and, thus, the suction of air out of the building.<br>• Air collector sheds: although the increase in the depth of the building has increased the number of sheds on the roof and, with this, the air inlet openings, there was a significant reduction in the use of external winds for all analyzed geometries. This is because not all of them became effective for collecting air, due to the wind shadow generated by the winds that touch the sheds. In cases with greater depth, the collection continued to occur only for the first shed, as in the case with three sheds, with no increase in airflow.<br>• Shed roof air extractors and collectors: better performance for aerodynamic geometries (an increase of 4% and 11% for G01 and G02, respectively, for air extractors sheds and 5% and 4% for G01 and G02, respectively, for air collectors sheds). However, it is noteworthy that the difference between the different geometries for air collectors sheds was insignificant. This reinforces that design parameters, such as the increase in the number of sheds and the depth of the building, influenced the performance of natural indoor ventilation more than just the change in the geometry of the sheds.<br>• Air extractor sheds performed better than collectors by approximately 10%. |

Figure 11 shows the use of external wind in the internal environment ($U/U_{ref}$) for the cases of group 3, for the shed roof air extractors and collectors. Table 8 gathers the design guidelines regarding the parameter opening, distance and alignment of the sheds, for the same situations.

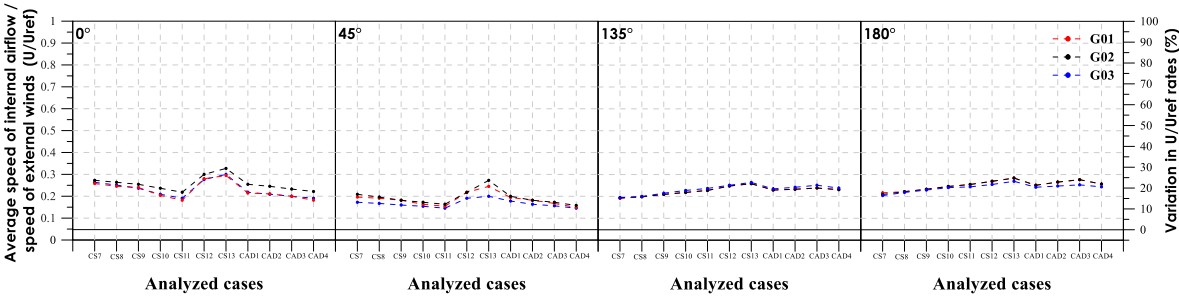

**Figure 11.** Percentage of utilization of the average speed of the indoor airflow as a function of the speed of the external winds ($U/U_{ref}$) for group 3: opening, distance and alignment of the sheds.

**Table 8.** Design guidelines for group 3: opening, distance and alignment of the sheds.

| Design Parameter 04: SIZE OF THE SHEDS AND DISTANCE BETWEEN THESE ELEMENTS | | |
|---|---|---|
| Air EXTRACTOR Sheds | Air COLLECTOR Sheds | Air EXTRACTORS and COLLECTORS |
| CS13 + G02<br>CS7 + G02 | CS11 + G01    CS11 + G02<br>CS13 + G01    CS13 + G02<br>CAD3 + G01    CAD3 + G02 | CS13 + G02 |

Based on Lukiantchuki et al. (2019) [22] , we have:

- Air extractor sheds: increasing the distance between the sheds, as well as increasing the distance with the misalignment of the sheds in two rows, reduced the total area of the openings of these devices, reducing the velocity of the indoor air significantly by 20%. Thus, it is recommended to: (1) keep sheds arranged closer to each other and aligned, due to the greater total area of the air outlet openings (CS7); (2) if the distance between the sheds increases, it is also recommended to increase the air outlet openings—CS13. It is noted that the total area of the air outlet openings was the parameter that exerted the greatest influence on the ventilation performance for these devices as air extractors.
- Air collector sheds: (1) greater spacing between these devices (CS11) that allow a slight increase in the use of the indoor airflow in 15%. In the case with the sheds close to each other, the air inlet occurred only through the first shed. As these devices were distanced from each other, the openings in the region posterior to the windward shed become more effective for the entry of air, allowing greater use of the winds internally; (2) the increase was more significant (around in 23%) when the increase in distance was combined with the increase in the openings of the sheds (CS13); (3) greater spacing between these devices and the misalignment of the sheds (CAD3). Although the inlet openings were reduced, they became more effective for collecting.
- Shed roof air extractors and collectors: CS13 is shown as the most viable case for situations of adaptation to changes in wind incidence angles.
- Despite the increase in the distance between the sheds having registered an increase in the indoor airflow for collection and a reduction for the air extractors, the performance was still superior for the air extractor sheds.
- The use of aerodynamic geometries is recommended. However, the performances of the different geometries (CR, G01 and G02) were similar (difference of around 2%), which would allow the use of orthogonal geometry, increasing the size of the outlet openings.

### 3.2. Design Recommendations Regarding the Velocity of External Winds

Figures 12–14 show the average air velocity in the internal environment for the cases of groups 1, 2 and 3, respectively, indicating the limits of acceptability of the indoor air velocity in relation to the velocity of the external winds (m/s), for the shed roof air extractors and collectors.

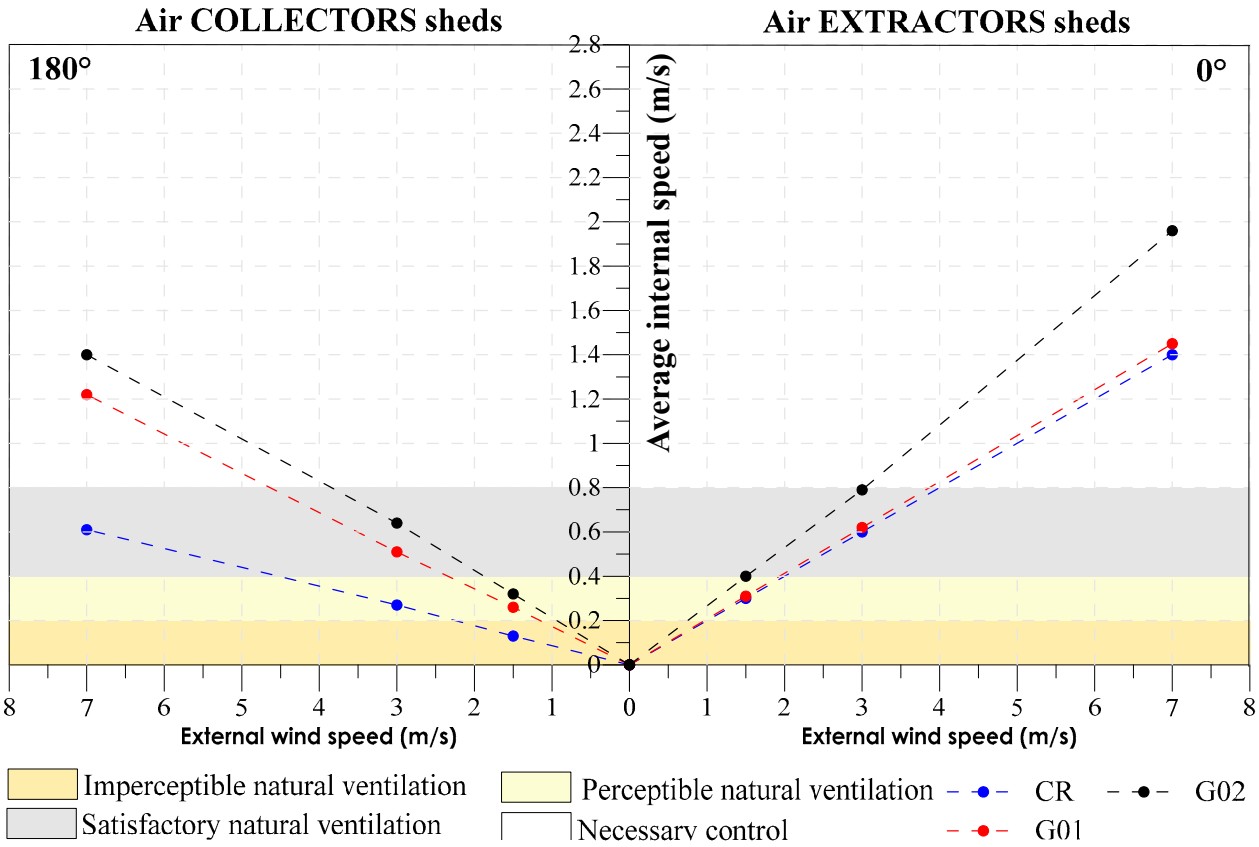

**Figure 12.** Internal average air velocity for group 1, indicating the limits of acceptability of the indoor air velocity in relation to the velocity of the external winds (m/s).

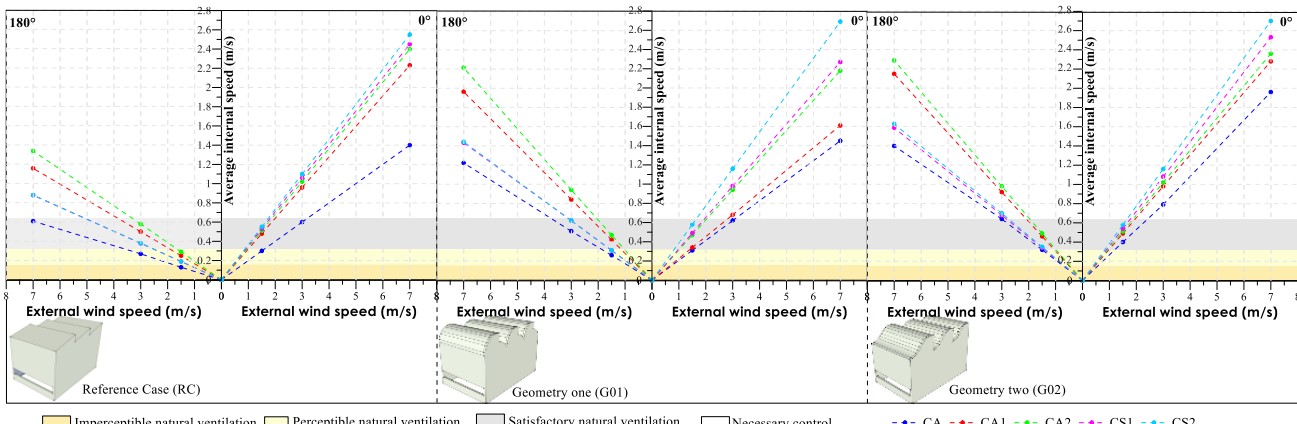

**Figure 13.** Internal average air velocity for group 2, indicating the limits of acceptability of the indoor air velocity in relation to the velocity of the external winds (m/s).

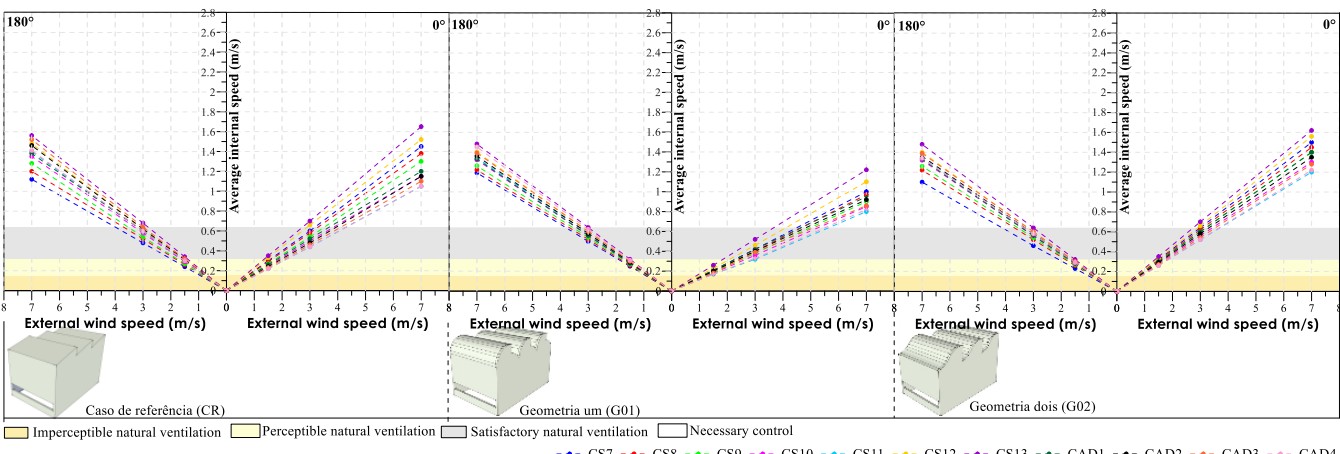

**Figure 14.** Internal average air velocity for group 1, indicating the limits of acceptability of the indoor air velocity in relation to the velocity of the external winds (m/s).

In general, extractor sheds reach internal speeds higher than collectors do. In the extraction situation, the G02 is suitable for both low and medium wind speeds. The CR and the G01, on the other hand, reach a satisfactory ventilation range only for medium speeds. For high-speed external winds, all geometries need strategies to control the winds in the internal environment. In relation to collection, for external winds of 1.5 m/s, none of the geometries had satisfactory internal ventilation. When the winds were 3.0 m/s, the G01 and G02 took advantage of the winds satisfactorily in the internal environment, unlike the CR, which needed strategies to increase the internal air velocity. This situation was reversed in regions with high speeds (7.0 m/s), because the CR became a good solution for internal natural ventilation while the G01 and the G02 needed to control the air inlet.

For extracting sheds, the increase in air in the internal spaces could be achieved both by increasing the air inlet and outlet openings, emphasizing that this increase was greater in the second case. Group 3 models, on the other hand, did not present themselves as viable alternatives to achieve satisfactory natural ventilation, because the air outlet openings have been reduced. Only the CS12 and the CS13 presented themselves as good solutions when the speeds are around 3.0 m/s, because the openings of the sheds increased, which allows a satisfactory internal ventilation.

As for the shed roof air collectors, the improvement of natural ventilation in internal environment can occur through the increase in the size of the air outlet openings and through the design modifications of the cases shown in group 3. This is because the increase in the distance between these devices, despite reducing the air inlet openings, made the posterior sheds more effective for collecting, which increased the indoor airflow.

Finally, it is important to emphasize that the choice of design strategies must be made according to the use of the building, as this influences the permissible limits of the velocity of the indoor airflow, such as schools that demand lower internal speeds than a gym, for example. In addition, it is noteworthy that the insertion of any control device at the entrance of the airflow, will modify the relation of the openings areas and, thus, the behavior of the proposed openings system. In relation to the architectural design, it is sought that the winds in the summer are incorporated to cool the environments as opposed to the winter winds that must be avoided to reduce the heat losses of the building to the outside. Therefore, design strategies that aim to incorporate natural ventilation in indoor environments should receive special attention in architectural design, often requiring design decisions that are different from conventional ones. Table 9 presents the results regarding the limits of acceptability of the indoor air velocity, in relation to the external wind velocity (m/s), for the shed roof air extractors and collectors in a summarized form.

**Table 9.** Summary of the design strategies according to the limits of acceptability of the indoor air velocity in relation to the velocity of the external winds (m/s).

| SHED ROOF AIR EXTRACTORS | | | |
| --- | --- | --- | --- |
| CLASSIFICATION | 1.5 m/s | 3.0 m/s | 7.0 m/s |
| IMPERCEPTIBLE | Group 01: CR and G01<br>Group 03: buildings with greater depth; increasing the distance between the sheds and the misalignment of the sheds. | —— | —— |
| PERCEPTIBLE | —— | —— | —— |
| SATISFACTORY | Group 01: G02<br>Group 02: increase in the size of the air inlet and outlet openings | Group 01: CR, G01, G02<br>Group 03: sheds close to each other; insert the sheds in 2 rows, without increasing the distance between them too much; if the distance between these devices increases, the air outlet openings must be increased, which increases internal ventilation, making it satisfactory for achieving comfort | —— |
| CONTROL | —— | Group 02: increase in the size of the air inlet and outlet openings | Group 01: all cases<br>Group 02: all cases<br>Group 03: all cases |
| SHED ROOF AIR COLLECTORS | | | |
| | 1.5 m/s | 3.0 m/s | 7.0 m/s |
| IMPERCEPTIBLE | Group 01: CR<br>Group 02: the increase in air inlet and outlet openings increased the velocity of the indoor airflow, but even so, the CR remained with unsatisfactory natural ventilation.<br>Group 03: buildings with greater depth; increase the distance between the sheds and the misalignment of the sheds | Group 01: CR<br>Group 02: increase in air inlet openings (sheds)<br>Group 03: all cases are enough to reduce the thermal load and provide thermal comfort for users. | —— |
| PERCEPTIBLE | Group 01: G01 and G02 | —— | —— |
| SATISFACTORY | Group 02: G01 and G02, with the increase in the air outlet opening users are comfortable | Group 01: G01 and G02<br>Group 02: increase in the size of the air outlet openings | —— |
| CONTROL | —— | —— | Group 01: all cases<br>Group 02: all cases<br>Group 03: all cases |

## 4. Conclusions

In general, this article highlights that there are differences in the design parameters for shed roof air extractors and collectors and that it is possible to optimize the use of natural ventilation through these devices. For air extractor sheds:

1. The use of aerodynamic geometries showed a better performance.

2. The increase in the air outlet openings presents a significantly increased the indoor airflow.

3. The use of buildings with fewer sheds and, therefore, less depth also showed a better performance for extractors. When buildings with greater depth are used, it is recommended to increase the distance between the sheds with the increase of the air outlet openings (sheds).

4. The isolated increase in the distance between the sheds or the increase in the distance with the misalignment of these devices significantly reduced the total area of the air outlet openings and, as a result, there was a lower speed of the indoor airflow. For these situations, it is better to keep the sheds closer together, due to the larger total area of the air outlet openings.

5. It should be noted that the total area of the air outlet openings was the design parameter that exerts the greatest influence on the performance of natural ventilation for air extractor sheds.

For air collector sheds

1. The use of aerodynamic geometries also performed better for natural ventilation.

2. The use of buildings with less depth allows a better performance, because the increase in the depth of the building increased the number of sheds in the roof, but not all sheds' openings were effective for air inlet.

3. As with extractor sheds, again, increasing the air outlet opening provided an increase in the speed of the indoor air. The increase in the sizes of the air inlet openings (sheds) caused a slight increase in the indoor airflow, but significantly lower than the increase in the air outlet opening (leeward).

4. The isolated increase in the sizes of the air inlet openings (sheds), without changing other parameters, such as the distance and the alignment between the sheds, did not make the posterior openings effective for inlet and thus the increase in the speed of the indoor air was not so significant.

In relation to the building with greater depth, it is recommended:

1. To increase the distance between these devices, because the larger spaces between the sheds made it possible to collect the air through the posterior sheds.

2. The performance was even greater if the increase in the distance between these devices was combined with the increase in the air inlet openings.

3. In addition, the increased distance between the sheds along with the change in alignment, made the air inlet openings more effective for the collection, improving performance.

4. For the collection, it was also emphasized that parameters such as the increase in the number of sheds and the depth of the building influenced the performance of natural ventilation more than just the change in the shape of the sheds.

As presented, the use of these devices, in each of these situations, required the adoption of specific design parameters that optimize the use of natural ventilation in indoor environments. However, it was noted that some configurations registered good performance for both situations, which is interesting when changes in the surroundings change the direction of the winds.

**Author Contributions:** Conceptualization, M.A.L., A.R.P.S. and F.M.d.S.; methodology, M.A.L., A.R.P.S. and F.M.d.S.; software CFD, M.A.L. and A.R.P.S.; formal analysis, M.A.L., A.R.P.S. and F.M.d.S.; resources, M.A.L., A.R.P.S. and R.M.C.; writing—original draft preparation, M.A.L., A.R.P.S., F.M.d.S. and R.M.C.; writing—review and editing, M.A.L., A.R.P.S., F.M.d.S. and R.M.C. All authors have read and agreed to the published version of the manuscript.

**Funding:** This research was funded by São Paulo Research Foundation (FAPESP), grant number 2011/11376-6 and Notice CCP AU 01/2021—PROAP/FAUUSP.

**Acknowledgments:** Grant #2011/11376-6, São Paulo Research Foundation (FAPESP), and Notice CCP AU 01/2021—PROAP/FAUUSP.

**Conflicts of Interest:** The authors declare no conflict of interest.

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
