# Peer review of "Shed Roof Air Extractors and Collectors: Design Guidelines for Natural Ventilation in Generic Models"

_2674-032X, doi:10.3390/wind3020011_

Round 1

Reviewer 1 Report

The paper is interesting and well written. The following basic clarifications need to be done.

U, Uref to be defined in line 239

Air velocity in table 4 is  at which point? how is it averaged over the room, over which volumes?

What is the external wind values used for simulation and what was the direction of flow?

The terms  shed, collector, extractor and other terms can be defined more clearly in the introduction for the reader to have easier understanding

What happens to the results and conclusions when a) there are nearby building obstructions b) when the wind speed is lower

Can the model be validated by an experimental study or a reference study can be modelled to validate the model? 

Author Response

The paper is interesting and well written. The following basic clarifications need to be done.

Thank you so much for carefully reviewing the article. We've improved the introduction and other suggested items.

U, Uref to be defined in line 239

Thanks. These parameters were better defined in the article.

Air velocity in table 4 is  at which point? how is it averaged over the room, over which volumes?

These velocities in Table 4 are adopted based on Cândido et al. As are the analysis parameters used based on another research were not detailed in this article. However, a highlight on this data was included in the search method.

What is the external wind values used for simulation and what was the direction of flow?

Thanks. This information is described in item 2.2 - Selection of Climatic Date of Materials and Method.

The terms  shed, collector, extractor and other terms can be defined more clearly in the introduction for the reader to have easier understanding

The description of the air extractors and collectors was rewritten, aiming to improve readers' understanding.

What happens to the results and conclusions when a) there are nearby building obstructions b) when the wind speed is lower

Thank you very much. The note is necessary and was discussed in the article. With this in mind, we inserted the simulations input data the external speed correction equation, generating the atmospheric boundary layer.

Can the model be validated by an experimental study or a reference study can be modelled to validate the model? 

Thank you very much. The validation process is very important and was conducted in the present research. However, as it was the focus of another article, we explained this in the method (item 2.3.1) and cited the article where the simulated models were validated in the wind tunnel.“The computational simulations were validated through wind tunnel tests in National Civil Engineering Laboratory – LNEC, Lisbon. The results were published in Lukiantchuki, et al., (2018) [17] and Lukiantchuki, et al., (2019) [27]”.

We highlight that the material item and method has been reviewed and some better points described.

Reviewer 2 Report

Dear Authors

Interesting article. Please check the following:

1. Natural ventilation is important SDGoals! and particularly in hot climates – there's no better way to passively cool down. It is therefore a topic of high interest and relevance.
2. This type of roof – shed is normally used in industrial buildings / gyms / swimming pools – I think the article intends to address single-family houses, isn´t it? – clarify this point Line 110, title
3. Make clear what distinguishes the work in this article from the works presented in 2016, 2017 and 2019
4. Is there no comparison between wind tunnel tests and computer simulations?
5. Table 4 – limits of acceptability of air speed – type of space: housing, service, industry?
6. Figs 10, 12 and 13 difficult to read

Author Response

  1. A ventilação natural é importante SDGoals! e particularmente em climas quentes – não há melhor maneira de se refrescar passivamente. É, portanto, um tema de grande interesse e relevância.

Thank you very much. We were very happy with this comment.

  1. Esse tipo de telhado – galpão é normalmente utilizado em prédios industriais/ginásios/piscinas – acho que o artigo pretende abordar casas unifamiliares, não é mesmo? – esclarecer este ponto Linha 110, título

Thank you very much. These devices are more used for industrial buildings. In Brazil, this device is widely used by Lelé in health care establishments. In our case, we do not limit any specific use, considering the analysis for generic models. However, we explain this better in the title and article.

  1. Deixar claro o que distingue o trabalho deste artigo dos trabalhos apresentados em 2016, 2017 e 2019

Thank you very much. An explanation was inserted in the last paragraph of the introduction.

  1. Não há comparação entre testes de túnel de vento e simulações de computador?

Thank you very much. The validation process is very important and was conducted in the present research. However, as it was the focus of another article, we explained this in the method (item 2.3.1) and cited the article where the simulated models were validated in the wind tunnel. “The computational simulations were validated through wind tunnel tests in National Civil Engineering Laboratory – LNEC, Lisbon. The results were published in Lukiantchuki, et al., (2018) [17] and Lukiantchuki, et al., (2019) [27]”.

  1. Tabela 4 – limites de aceitabilidade da velocidade do ar – tipo de espaço: habitação, serviço, indústria?

These velocities in Table 4 are adopted based on Cândido et al. As are the analysis parameters used based on another research were not detailed in this article, but were analyzed for office buildings.

  1. Figuras 10, 12 e 13 difíceis de ler

The figures have been changed to improve visualization.

 We highlight that the material item and method has been reviewed and some better points described.

Reviewer 3 Report

The paper presents one type of roof air extractors and collectors  and discusses the differences in the design parameters combination also considering the type of building on which are posed. The results are obtained with the help of a simulator. The analysis is well presented and the paper is of great interest for whom considers this technique. It would be interesting report also different architecture of air extractors and collectors and report some pros and cons of forced air conditioning.

At line 174 the word "simulations" is repeated.

Maybe, some tables can be considered pictures

Author Response

The paper presents one type of roof air extractors and collectors  and discusses the differences in the design parameters combination also considering the type of building on which are posed. The results are obtained with the help of a simulator. The analysis is well presented and the paper is of great interest for whom considers this technique. It would be interesting report also different architecture of air extractors and collectors and report some pros and cons of forced air conditioning.

Thank you very much for the correction and for the comments. We are very happy with this comment. In the introduction we mention the works of the architect Lelé who uses different types of sheds in his buildings.

At line 174 the word "simulations" is repeated.

Thank you very much. The word has been omitted.

Maybe, some tables can be considered pictures

Thanks. We ended up keeping them as tables, according to the article's template.

Reviewer 4 Report

1.       Considering the paper revolves around the natural cooling of houses and building especially in Brazilian government, I suggest authors to add few recent governmental policies favouring natural ventilation (if any).

2.       The different building / shed shapes can be made as a table depicting the author name, type of shed, their discrete advantages , disadvantages etc. This will help the readers not only understand more about the various types of roofs but also enhances the citation for the paper inviting more reads.

3.       Line No: 79-80, when we are citing more than 2 papers, rather than depicting it like [20,21,…26], it can be mentioned as [20-26]. Please consider.

4.       Line No:86-102, the points can be made individually if the author(s) wish, numbering them in a paragraph may mislead the readers. Please rectify.

5.       Table 2 and Table 3 needs to be categorized in a way it is kind to the readers. Atleast, the table description should convey the details related to it. The table 2 and table 3 can be described in the text for the benefit of readers.

6.       Why unstructured mesh scheme was preferred over the structured mesh which could have rendered less computational time. Is it random or intentionally chosen.

7.       Details regarding the simulation of the ABL profile were not discussed in the manuscript. Similar to that no measurement datas or CFD contours has been shown to prove the same. Kindly clarify. What is the y+ value of the grid considered in this present study.

8.       Section 2.3.3., the boundary conditions can be depicted as a table for easier understanding.

9.       Why convergence criterion is limited to 10-4, and the minimum iteration at 1000. Is it based on some literature or randomly chosen.

10.   Results needs to be explained clearly, the reviewer understands the qualitative flow pattern suggested by the author(s), but quantitative information is still lack. Figure 7 lacks description quantitatively. Similar to Figure 8 and 9 also. Why the pressure and velocity contours are not used by the author(s) to substantiate the same theory.

11.   Line No:328-329; the author(s) claim that “As for the shed roof air collectors, the achievement of thermal comfort can occur through the increase in the size of the air outlet openings” – Kindly please prove that the thermal comfort occurs with the increase in the size of the air outlet opening with the help of suitable contour plots or graphs.

12.   From Table 9, it is again understood that the computational methodology adopted by the author(s) are not well planned. Why some of the cases are not analyzed? What is the reason behind the same, are they intentionally left out or because of the computational effort and time constraint they are neglected.

13.   Conclusion can be made crisp and clear with highlighted points rather than a full page long paragraph.

Author Response

  1. Considerando que o artigo gira em torno do resfriamento natural de casas e edifícios, especialmente no governo brasileiro, sugiro aos autores que acrescentem algumas políticas governamentais recentes que favoreçam a ventilação natural (se houver).

Thank you so much for the comment. However, unfortunately we do not have recent government policies that favor the use of natural ventilation. Thus, we do not enter this information in the article.

  1. Linha nº: 79-80, quando citamos mais de 2 artigos, em vez de descrevê-lo como [20,21,…26], pode ser mencionado como [20-26]. Por favor considere.

Thank you very much. This has been revised throughout the article.

  1. Linha No:86-102, os pontos podem ser feitos individualmente se o(s) autor(es) desejarem, numerá-los em um parágrafo pode confundir os leitores. Por favor, corrija.

Thank you very much. This change was made.

  1. A Tabela 2 e a Tabela 3 precisam ser categorizadas de forma a agradar aos leitores. Pelo menos, a descrição da tabela deve transmitir os detalhes relacionados a ela. A tabela 2 e a tabela 3 podem ser descritas no texto para benefício dos leitores.

Thank you very much. This change was made.

  1. Por que o esquema de malha não estruturada foi preferido em relação à malha estruturada, que poderia render menos tempo computacional. É aleatório ou intencionalmente escolhido.

The mesh type as well as the dimensions of the mesh elements were chosen based on mesh sensitivity tests. These tests were presented in another publication (Lukiantchuku (2018) - Reference 30) and as it was not the focus of this article these results were referenced and were not presented in this article.

  1. Detalhes sobre a simulação do perfil ABL não foram discutidos no manuscrito. Semelhante a isso, nenhum dado de medição ou contornos CFD foi mostrado para provar o mesmo. Gentileza esclarecer. Qual é o valor y+ da malha considerada neste estudo.

Thank you very much. An explanation has been inserted, as well as the image of Atmospheric Boundary Layer.

  1. Seção 2.3.3., as condições de contorno podem ser representadas como uma tabela para facilitar a compreensão.

Thanks for the comment. But we think the text is explained and the insertion of the table along with the text could be repetitive.

  1. Por que o critério de convergência é limitado a 10-4 e a iteração mínima em 1000. É baseado em alguma literatura ou escolhido aleatoriamente?

O critério de convergência 10-4 é definido de acordo com a literatura especializada. Inúmeros trabalhos adotam esse valor. The Minimum Number of Itertations was chosen at random, as it does not have a recommended value to be adopted in the specialized literature. We emphasize that all simulations have been converged.

  1. Os resultados precisam ser explicados com clareza, o revisor entende o padrão de fluxo qualitativo sugerido pelo(s) autor(es), mas ainda faltam informações quantitativas. A Figura 7 carece de descrição quantitativa. Semelhante à Figura 8 e 9 também. Por que os contornos de pressão e velocidade não são usados pelo(s) autor(es) para fundamentar a mesma teoria.

Thank you so much for the analysis. The results were complemented, deepening the quantitative discussions presented in the graphs.

  1. Linha No:328-329; o(s) autor(es) afirma(m) que “No caso dos coletores de ar da cobertura shed, o alcance do conforto térmico pode ocorrer através do aumento do tamanho das aberturas de saída de ar” – Por favor, comprove que o conforto térmico ocorre com o aumento do tamanho da abertura de saída de ar com a ajuda de plotagens ou gráficos de contorno adequados.

Thanks for the comment. As thermal comfort was not evaluated, we replaced the sentence considering that it would have an improvement in natural ventilation in the internal environment, which can be a factor in improving users' thermal comfort in hot and moist climate regions.

  1. Da Tabela 9, novamente se entende que a metodologia computacional adotada pelo(s) autor(es) não é bem planejada. Por que alguns dos casos não são analisados? Qual é a razão por trás disso, eles são deixados de fora intencionalmente ou por causa do esforço computacional e da restrição de tempo, eles são negligenciados.

All cases were analyzed and, according to their performance, were classified in the table to summarize the results found. I believe I have occurred an interpretation error. “In the analyzed cases” refers to that none of the cases analyzed fits this category of the air velocity analyzed. For example, when the external air speed was 3.0m/s, no case had a low and imperceptible air flow speed by users. Anyway, I changed the table nomam to do not generate interpretation problems.

  1. A conclusão pode ser nítida e clara com pontos destacados, em vez de um parágrafo de página inteira.

Thank you so much for the considerations. The conclusion was reformulated.